# Offline-to-Online Reinforcement Learning with Classifier-Free Diffusion Generation

Xiao Huang [* 1]  Xu Liu [* 1]  Enze Zhang [1]  Tong Yu [2]  Shuai Li [1]

## Abstract

Offline-to-online Reinforcement Learning (O2O RL) aims to perform online fine-tuning on an offline pre-trained policy to minimize costly online interactions. Existing work uses offline datasets to generate data that conform to the online data distribution for data augmentation. However, generated data still exhibits a gap with the online data, limiting overall performance. To address this issue, we propose a new data augmentation approach, **C**lassifier-**F**ree **D**iffusion **G**eneration (CFDG). Without introducing additional classifier training overhead, CFDG leverages classifier-free guidance diffusion to significantly enhance the generation quality of offline and online data with different distributions. Additionally, it employs a reweighting method to enable more generated data to align with the online data, enhancing performance while maintaining the agent's stability. Experimental results show that CFDG outperforms replaying the two data types or using a standard diffusion model to generate new data. Our method is versatile and can be integrated with existing offline-to-online RL algorithms. By implementing CFDG to popular methods IQL, PEX and APL, we achieve a notable 15% average improvement in empirical performance on the D4RL benchmark such as MuJoCo and AntMaze.

## 1. Introduction

Traditionally, Reinforcement Learning (RL) (Haarnoja et al., 2018) is considered a paradigm for online learning, where agents learn from online interactions with the environment. Due to costly online interactions in some real-world applications, offline RL (Levine et al., 2020) is proposed where

---
[*]Equal contribution [1]Shanghai Jiao Tong University, Shanghai, China [2]Adobe Research, California, United States. Correspondence to: Shuai Li <shuaili8@sjtu.edu.cn>.

*Proceedings of the 42nd International Conference on Machine Learning*, Vancouver, Canada. PMLR 267, 2025. Copyright 2025 by the author(s).

agents learn from a static dataset pre-collected by arbitrary policies. Current research in offline RL focuses primarily on addressing the challenge of distribution mismatch or out-of-distribution (OOD) actions through the implementation of a pessimistic update scheme (Kumar et al., 2020) or in combination with imitation learning (Kumar et al., 2019). However, when dealing with a fixed and suboptimal dataset, it becomes exceedingly challenging for offline RL to attain the optimal policy (Kidambi et al., 2020).

Some recent work addresses the above issues by employing an offline-to-online setting. Such methods (Lee et al., 2022; Nair et al., 2020) focus on pre-training a policy using the offline dataset and fine-tuning the policy through further online interactions. O2O RL aims to fine-tune a pre-trained policy using limited online interactions to achieve the optimal policy. To utilize the offline dataset, some studies directly replayed samples in the online phase (Lee et al., 2022), leading to performance improvements. Nevertheless, this approach neglects the distribution shift issue, as the data distribution in the offline dataset may differ from that induced by the current policy. To address the distribution shift problem, existing work (Zheng et al., 2023) suggests that their different characteristics require separate updating strategies for online and offline data, respectively. In particular, offline data can deter agents from prematurely converging to suboptimal policies due to the diversity of available data, while online data can contribute to training stability and accelerate convergence (Nair et al., 2020; Thrun & Littman, 2000).

In addition to fully utilizing offline and online data, there are also works that leverage generative models for data augmentation to provide diverse samples for the agent's training. Prior work has considered upsampling online data with VAEs or GANs (Huang et al., 2017; Imre, 2021). To fully leverage offline data, Energy-guided Diffusion Sampling (EDIS) (Liu et al., 2024) generates data aligned with the online policy based on the offline dataset to address the issue of distribution shift. In addition to using a diffusion model to generate data, it formulates three distinct energy functions to guide the diffusion sampling process, ensuring alignment with the online policy. Intuitively, online data is more closely aligned with the current online policy. The limitation of previous works lies in using of-

fline data for data augmentation, resulting in generated data that still exhibits significant discrepancies from the online policy. Although it achieves good results, adding multiple new models introduces additional training overhead and time costs, and the entire guidance process becomes quite complex. Consequently, a fundamental question arises: *Can we use a simpler approach to guide the model in generating higher-quality data?*

To address this issue, we revisit the relationship between offline data and online data and conduct a distributional analysis for both types of data and the generated data. Given the completely distinct characteristics of offline and online data, we perform data augmentation for both simultaneously. To simplify the model, we utilize a classifier-free guidance diffusion model for data augmentation. This approach serves two purposes: first, it treats offline and online data as two labeled categories, enabling simultaneous sampling of both types with a single training process, thereby reducing time costs; second, it avoids using an additional pre-trained classifier, allowing data augmentation to adapt to varying data distributions in different RL tasks. After data generation, we use a reweighting method to prioritize data that is more aligned with the online policy for agent training, aiming to improve performance. Experimental results show that our CFDG method not only enhances the quality of generated samples compared to existing model-based methods but also significantly improves the performance of O2O RL. Our contributions are summarized below:

- We analyzed the distributions of offline data, online data, and the generated data in O2O RL and found that the current method still generates data that is not sufficiently aligned with the online policy.

- By treating offline data and online data as two distinct labels, we introduce classifier-free guidance to direct the diffusion model in generating both types of data. We perform reweighting on the generated data to make it more aligned with the online policy.

- We conducted experiments on the Locomotion and AntMaze tasks, demonstrating that our data augmentation method significantly improves multiple O2O RL algorithms, including IQL, PEX, and APL. Furthermore, our approach outperforms the existing data augmentation method EDIS.

## 2. Preliminaries

We represent the environment as a Markov decision process (MDP) defined by a tuple $(\mathcal{S}, \mathcal{A}, P, R, \rho_0, \gamma)$, where $\mathcal{S}$ is the state space, $\mathcal{A}$ is the action space, $P(s' \mid s, a)$ is the transition distribution, $\rho_0$ is the initial state distribution, $R(s, a)$ is the reward function and $\gamma \in (0, 1)$ is the discount factor. The objective of RL agent is to find a policy

$\pi(a \mid s)$ that maximizes the expected cumulative return $\mathbb{E}_\pi[\sum_{t=0}^{\infty} \gamma^t r_{t+1}]$.

### 2.1. Offline Reinforcement Learning

In Offline RL, the agent can only access a static dataset $\mathcal{D}$ collected by a behavior policy $\pi_\beta(a \mid s)$. Offline RL approaches can take advantage of the offline dataset to train a critic network (Q-function) $Q_\theta^\pi(s, a)$ with parameters $\theta$, which estimates the long-term discounted reward achieved by executing action $a$ in state $s$ and following the policy $\pi$. The critic network can be trained using the following temporal difference (TD) learning objective:

$$\mathcal{L}_Q(\theta) = \mathbb{E}_{(s,a,r,s') \sim \mathcal{D}} \left[ \left( r + \gamma Q_{\hat{\theta}}\left(s', \pi_\phi\left(s'\right)\right) - Q_\theta(s, a) \right)^2 \right],$$
(1)

where $\hat{\theta}$ denotes the target value network for stabilizing the learning process. Since $\pi_\phi(s')$ is potentially out of the distribution, $Q_\theta$ could give an incorrect value, resulting in suboptimal policies. To mitigate the well-known extrapolation error in value networks for OOD actions (Fujimoto et al., 2019; Kumar et al., 2020), offline RL methods typically constrain the policy to perform actions close to the dataset through policy constraint (Fujimoto et al., 2019; Fujimoto & Gu, 2021; Kumar et al., 2019), value regularization (Kumar et al., 2020; An et al., 2021), etc. One representative offline RL method is CQL (Kumar et al., 2020). CQL adds a conservative regularizer $\mathcal{R}(\theta)$ to prevent overestimation in the Q-values for OOD actions by minimizing the Q-values under the policy $\pi$. The training objective of CQL is given by

$$\min \lambda \underbrace{\left( \mathbb{E}_{s \sim \mathcal{D}, a \sim \pi}\left[Q_\theta(s, a)\right] - \mathbb{E}_{s, a \sim \mathcal{D}}\left[Q_\theta(s, a)\right] \right)}_{\text{Conservative regularizer } \mathcal{R}(\theta)} + \frac{1}{2}\mathcal{L}_Q(\theta),$$
(2)

where $\lambda$ balances the standard policy improvement loss and conservative regularization.

### 2.2. Offline-to-online Reinforcement Learning

Building upon the concepts of offline RL, offline-to-online RL aims at enhancing performance by fine-tuning pre-trained offline policy, which contains two phases: (i) *offline pre-training*, where offline datasets are used to pre-train the policy, and (ii) *online fine-tuning*, where online interactions are used to refine the pre-trained policy.

Currently, in O2O RL algorithms, there are two paradigms for utilizing offline data and online data. An approach is to set the ratio of online data to offline data at 1:1, ensuring each batch contains an equal split of online and offline data,

such as PEX (Zhang et al., 2023) and Cal-QL (Nakamoto et al., 2024). Another approach utilizes an online-offline replay buffer (OORB) in APL (Zheng et al., 2023) and SUNG (Guo et al., 2023), with each batch having a probability $p$ of containing online data and a probability $1 - p$ of containing offline data. As mentioned in Equation (2), $\lambda$ is a trade-off coefficient and decides whether we use the regularizer. When data is sampled from the online buffer, $\lambda$ is set to 0, otherwise 1. Formally, this strategy can be explained below:

$$\lambda \leftarrow \begin{cases} 0 & \text{if } (\mathbf{s}, \mathbf{a}) \sim \text{online buffer} \\ 1 & \text{otherwise.} \end{cases} \quad (3)$$

### 2.3. Diffusion Models

Diffusion models (Ho et al., 2020) are a class of generative models inspired by non-equilibrium thermodynamics that learn to iteratively reverse a forward noising process and generate samples from noise. Diffusion models define a probability distribution $p_\theta(x_0)$ for the observed data $x_0$ by marginalizing over the latent variables $x_1, \ldots, x_T$, where $p_\theta(x_0) := \int p_\theta(x_{0:T}) dx_{1:T}$. A forward diffusion chain gradually adds noise to the data $x_0 \sim q(x_0)$ in $T$ steps with a pre-defined variance schedule $\beta_i$, expressed as

$$\begin{aligned} q(x_{1:T} \mid x_0) &:= \prod_{t=1}^{T} q(x_t \mid x_{t-1}), \\ q(x_t \mid x_{t-1}) &:= \mathcal{N}(x_t; \sqrt{1 - \beta_t} x_{t-1}, \beta_t I). \end{aligned} \quad (4)$$

A reverse diffusion chain, constructed as $p_\theta(x_{0:T}) := \mathcal{N}(x_T; \mathbf{0}, I) \prod_{t=1}^{T} p_\theta(x_{t-1} \mid x_t)$, is then optimized by maximizing the evidence lower bound defined as $\mathbb{E}_q[\ln \frac{p_\theta(x_{0:T})}{q(x_{1:T} \mid x_0)}]$ (Blei et al., 2017). After training, sampling from the diffusion model involves sampling $x_T \sim p(x_T)$ and running the reverse diffusion chain to go from $t = T$ to $t = 0$.

### 2.4. Classifier-free Guidance

In practical scenarios, a growing demand exists to condition the generation on a label $c$. For example, diffusion models can generate images consistent with input prompts in image synthesis. To address this requirement, classifier guidance (Dhariwal & Nichol, 2021) incorporates an auxiliary classifier $p_\phi(c|x_t)$ to guide the sampling in each reverse denoising step, thereby increasing the likelihood of $c$ given $x_t$. While this method has demonstrated some performance improvements, training a robust classifier for all reverse steps, particularly for the highly noisy input at the initial step, poses a significant challenge and incurs additional training costs.

To avoid training a separate classifier model, classifier-free guidance (Ho & Salimans, 2021) takes $c$ as another input of the denoising neural network to model the conditional diffusion score, i.e., $\epsilon_\theta(x_t, c, t) \approx -\sigma_t \nabla_{x_t} \log p(x_t|c)$ while the unconditional score $\epsilon_\theta(x_t, t)$ is jointly estimated by randomly dropping the text prompt with a certain probability at each training iteration. Then the gradients for the classifier $p_\phi(c|x_t)$ can be estimated as:

$$\begin{aligned} \nabla_{x_t} \log p(c|x_t) &= \nabla_{x_t} \log p_\theta(x_t|y) - \nabla_{x_t} \log p_\theta(x_t) \\ &= -\frac{1}{\sigma_t}(\epsilon_\theta(x_t, c, t) - \epsilon_\theta(x_t, t)). \end{aligned} \quad (5)$$

Then the corresponding diffusion score can be derived as:

$$\hat{\epsilon}_\theta(x_t.c, t) = \epsilon_\theta(x_t, t) + w(\epsilon_\theta(x_t, c, t) - \epsilon_\theta(x_t, t)), \quad (6)$$

where $w$ is set as a global scalar parameter to control the guidance degree of the condition.

## 3. Classifier-free Diffusion Generation

In order to perform data augmentation for different types of data, in this section, we first visualize the data distributions of offline and online data in O2O RL and analyze the data generated by EDIS (Liu et al., 2024). To ensure the generated data aligns better with online data, we chose to use classifier-free guidance to direct the diffusion model in generating both online and offline data. This approach eliminates the need for a pre-trained classifier before sampling, reducing training costs and making it more adaptable to different RL tasks than other guidance methods.

### 3.1. Data Distribution Analysis

To effectively perform data augmentation in O2O RL, it is essential to conduct a detailed analysis of the distributions of both offline and online data. We conducted experiments on EDIS and simultaneously analyzed the data generated by it. We employed t-SNE (Van der Maaten & Hinton, 2008), a dimensionality reduction technique designed to visualize high-dimensional data by mapping it into a low-dimensional space while preserving local structures. This method was used to analyze three types of data: the offline dataset, the online data collected during the fine-tuning process, and the generated data from the diffusion model.

As shown in the left panel of Figure 1, the offline data is more evenly distributed, while the online data is more dispersed. The generated data primarily learns from the offline data and is adjusted towards the current online policy using energy guidance. As a result, the generated data retains most of the characteristics of the offline data while also exhibiting some similarities to the online data. This is why EDIS opts

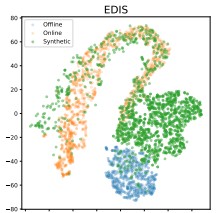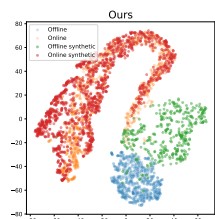

Figure 1: The t-SNE visualization of offline, online, and generated data. The left plot shows generated data from EDIS, and the right plot shows data from CFDG (Ours).

to replace the offline data with the generated data during the agent training process, using the generated data in conjunction with the online data. Although EDIS allows offline data to align with online data to some extent, it overlooks the fact that online data itself can also be augmented. Given the limited number of online data samples, augmenting online data can better adapt to the current policy and provide more samples to strengthen the policy's performance. In the Section 4.2, we also found that direct use of online data for generation yields better results than EDIS.

Based on the above analysis, directly augmenting online data is more effective than augmenting offline data. However, we also aim to maximize the utilization of both online and offline data, addressing the limited quantity of online data while ensuring that both types of data are better aligned with the current policy. Then, whether to augment both types of data naturally led us to consider using conditional diffusion. To better adapt both types of data to the current policy changes, we use classifier guidance to steer the diffusion model in generating new data that aligns with the changing data distribution.

### 3.2. Classifier-free Guidance Sampling

Classifier-free guidance (Ho & Salimans, 2021) does not require an additional classifier but instead trains the model using both conditional and unconditional inputs based on labels. For the classifier guidance method (Dhariwal & Nichol, 2021), the classifier is trained using noise-corrupted data produced by the forward process of the conditional diffusion model. Consequently, training an additional classifier can be difficult, especially when a significant amount of noise is added to the clean data. Classifier-free guidance can solve the above issues and eliminate the need for a pre-trained classifier before sampling, reducing training costs and making it more adaptable to different tasks than the classifier guidance method.

We built on the original code implementation of classifier-free guidance, integrating it into the Elucidated Diffusion Model (Karras et al., 2022). We chose to train an unconditional diffusion model $p_\theta(\mathbf{z})$ parameterized through a

score estimator $\boldsymbol{\epsilon}_\theta(\mathbf{z}_\lambda)$ together with a conditional model $p_\theta(\mathbf{z}|\mathbf{c})$ parameterized through $\boldsymbol{\epsilon}_\theta(\mathbf{z}_\lambda, \mathbf{c})$. We use a single neural network to parameterize both models, where for the unconditional model we can simply input a null token $\varnothing$ for the class identifier $\mathbf{c}$ when predicting the score, i.e. $\boldsymbol{\epsilon}_\theta(\mathbf{z}_\lambda) = \boldsymbol{\epsilon}_\theta(\mathbf{z}_\lambda, \mathbf{c} = \varnothing)$. We jointly train the unconditional and conditional models simply by randomly setting $\mathbf{c}$ to the unconditional class identifier $\varnothing$ with some probability $p_{\text{uncond}}$, set as a hyperparameter. We then perform sampling using the following linear combination of conditional and unconditional score estimates:

$$\tilde{\boldsymbol{\epsilon}}_\theta(\mathbf{z}_\lambda, \mathbf{c}) = (1+w)\boldsymbol{\epsilon}_\theta(\mathbf{z}_\lambda, \mathbf{c}) - w\boldsymbol{\epsilon}_\theta(\mathbf{z}_\lambda), \qquad (7)$$

where $w$ is a parameter that controls the degree of the classifier guidance. $\tilde{\boldsymbol{\epsilon}}_\theta$ is constructed from score estimates that are non-conservative vector fields due to the use of unconstrained neural networks, so there in general cannot exist a scalar potential such as a classifier log likelihood for which $\tilde{\boldsymbol{\epsilon}}_\theta$ is the classifier-guided score.

---

**Algorithm 1** Classifier-free guidance sampling in O2O RL. Our additions are highlighted in **blue**.

---

**Input:** Offline phase loss function $\{\mathcal{L}_{\text{offline}}^{Q_\theta}, \mathcal{L}_{\text{offline}}^{\pi_\phi}\}$, online phase loss function $\{\mathcal{L}_{\text{online}}^{Q_\theta}, \mathcal{L}_{\text{online}}^{\pi_\phi}\}$, classifier-guidance diffusion model $M$.
**Initialize:** $\theta, \phi$, online buffer $D_{\text{on}}$, offline buffer $D_{\text{off}}$, offline synthetic buffer $D_{\text{off\_syn}}$, online synthetic buffer $D_{\text{on\_syn}}$
**while** in offline training phase **do**
    Offline policy training using batches from the offline replay buffer $D_{\text{off}}$
    $\theta \leftarrow \theta - \lambda_Q \nabla_\theta \mathcal{L}_{\text{offline}}^{Q_\theta}(\theta), \phi \leftarrow \phi - \lambda_\pi \nabla_\phi \mathcal{L}_{\text{offline}}^{\pi_\phi}(\phi)$
**end while**
**while** in online training phase **do**
    **for** each environment step **do**
        $D_{\text{on}} \leftarrow D_{\text{on}} \cup \{(s, a, s', r)\}$
    **end for**
    **if** step meets $M$ update frequency **then**
        Update the conditional diffusion model $M$ with samples from $D_{\text{off}}$ and $D_{\text{on}}$
        Generate offline samples from $M$ and add them to $D_{\text{off\_syn}}$
        Generate online samples from $M$ and add them to $D_{\text{on\_syn}}$
    **end if**
    **for** each gradient step **do**
        Sample data from $D_{\text{on}} \cup D_{\text{off}} \cup D_{\text{off\_syn}} \cup D_{\text{on\_syn}}$
        $\theta \leftarrow \theta - \lambda_Q \nabla_\theta \mathcal{L}_{\text{online}}^{Q_\theta}(\theta), \phi \leftarrow \phi - \lambda_\pi \nabla_\phi \mathcal{L}_{\text{online}}^{\pi_\phi}(\phi)$
    **end for**
**end while**

---

In the O2O RL setting, we use the classifier-free diffusion

model to augment both online data and offline data during the online phase. Algorithm 1 describes the classifier-free guidance sampling process in detail. To reduce the high time cost of constructing the diffusion model and generating data, we set an update frequency to perform data augmentation using the diffusion model at regular intervals. During the online training phase, we update our diffusion model $M$ using samples from the offline buffer $D_{\text{off}}$ and online buffer $D_{\text{on}}$. Then the diffusion model $M$ can sample synthetic data and add them to the synthetic buffer $D_{\text{off\_syn}}$ and $D_{\text{on\_syn}}$. After data generation, we can sample batches from the four buffers mentioned. To make the most of different types of data, we need to design specific data utilization methods tailored to different O2O RL algorithms.

### 3.3. Data Reweighting

O2O RL algorithms usually combine the offline data and online data with a ratio during the training process. Since the addition of synthetic data, another hyperparameter, the synthetic data ratio $r$ is also required in our framework. In our data augmentation method, synthetic data are sampled from $D_{\text{off\_syn}}$ and $D_{\text{on\_syn}}$. To better align with the online policy, we adjust the sampling ratio and reweight the data, increasing the weight of the online synthetic data.

We first treat the online synthetic data and offline synthetic data as a whole and define how they should be used in relation to the offline data and online data. According to the two data usage methods mentioned in Section 2.2, we design two new data usage methods for synthetic data. For the first method, which combines 50% online data and 50% offline data, we use a simple method that concatenates synthetic data with sampled online data and offline data. The synthetic data ratio $r$ represents the proportion of synthetic data in each batch. Each batch is a concatenation of online data, offline data, and synthetic data.

For the second method for utilizing data, online data and offline data are sampled from OORB following a Bernoulli distribution. With a probability $p$, data is sampled from the online buffer, and with probability $1 - p$, they are sampled from the offline buffer. Besides, when data is sampled from the online buffer, $\lambda$ is set to 0, otherwise 1, as Equation 3 shows. Since we add synthetic data into the framework, we use a simple method where synthetic data can be seen as part of online data or offline data and be used for training as the online data or offline data does. The synthetic data ratio $r$ is also used to represent the proportion of synthetic data in each batch. Each batch is a concatenation of online data and synthetic data or offline data and synthetic data.

For the ratio of the two types of synthetic data, since online synthetic data aligns better with the current online policy, we choose to increase the weight of online synthetic data. This ensures that the entire synthetic data set supports better

exploration for the agent. In the right plot of Figure 1, we also show the data distribution generated by our CFDG method. By reweighting and increasing the weight of online synthetic data, we enable the agent to perform sufficient exploration. Compared to EDIS, our data aligns better with the current online policy.

## 4. Experiments

In this section, we show the efficiency of our CFDG method through empirical validation. Section 4.1 commence by showcasing its excellent performance on the D4RL benchmark (Fu et al., 2020) and also shows generalizability and statistical improvements on baselines like IQL (Kostrikov et al., 2021), PEX (Zhang et al., 2023) and APL (Zheng et al., 2023). Section 4.2 compares our method, CFDG, with other model-based approaches such as SynthER (Lu et al., 2024) and EDIS (Liu et al., 2024), highlighting the superiority of our conditional diffusion model over other diffusion models. In Section 4.3, we perform an ablation study to examine two key components of our method, generating online data and generating offline data using CFDG. Both are shown to effectively improve the performance of the algorithm.

### 4.1. Offline-to-online RL Experiments

**Datasets** Our method is mainly validated on two D4RL (Fu et al., 2020) benchmarks: Locomotion and AntMaze, which are used by IQL (Kostrikov et al., 2021) and PEX (Zhang et al., 2023). Locomotion includes diverse environmental datasets collected by varying quality policies. We assess algorithms on hopper, halfcheetah, and walker2d environment datasets, each with four quality levels. AntMaze tasks involve guiding an ant-like robot in mazes of three sizes (umaze, medium, large), each with two different goal location datasets. We focus on the two larger size mazes (medium, large), as opposed to using the umaze, which leaves little room for further improvement). The evaluation environments are listed in Table 1's first column. Additional experiments on the Adroit benchmark are shown in Appendix A.1.

**Baselines** We consider the following baselines: (i) **IQL** (Kostrikov et al., 2021) learns a value network to match the expectile of the critic network to address out-of-distribution problems. (ii) **PEX** (Zhang et al., 2023) freezes the pre-training policy and introduces policy expansion to enhance exploration. (iii) **APL** (Zheng et al., 2023) leverages the distinct advantages of offline and online data for adaptive constraints. These three are SOTA O2O RL algorithms and cover two paradigms of data utilization. , achieving excellent performance on the D4RL benchmark. According to Section 2.2, IQL and PEX, they follow the

Table 1: **Enhanced performance achieved by CFDG after online fine-tuning on the Loco-motion and AntMaze tasks.** We evaluate the normalized scores of standard base algorithms (including IQL (Kostrikov et al., 2021), PEX (Zhang et al., 2023) and APL (Zheng et al., 2023), denoted as "Base") in comparison to the base algorithms augmented with CFDG (referred to as "Ours"). All results are assessed across 5 random seeds. The superior scores are highlighted in blue .

| Dataset[1] | IQL | | PEX | | APL | |
|---|---|---|---|---|---|---|
| | Base | Ours | Base | Ours | Base | Ours |
| halfcheetah-r-v2 | 53±6 | 65 ± 3 | 78±2 | 81 ± 7 | 93±8 | 103 ± 7 |
| halfcheetah-mr-v2 | 54±0 | 65 ± 2 | 68±3 | 83 ± 3 | 76±40 | 96 ± 2 |
| halfcheetah-m-v2 | 69±2 | 75 ± 1 | 78±5 | 87 ± 3 | 77±39 | 86 ± 28 |
| halfcheetah-me-v2 | 95 ± 1 | 93±1 | 90±3 | 93 ± 0 | 96±3 | 98 ± 2 |
| hopper-r-v2 | 16 ± 13 | 10±1 | 8±0 | 8 ± 0 | 51 ± 30 | 30±40 |
| hopper-mr-v2 | 66±33 | 86 ± 30 | 66±25 | 83 ± 20 | 88±29 | 100 ± 15 |
| hopper-m-v2 | 93±6 | 97 ± 7 | 91±30 | 100 ± 7 | 103 ± 2 | 99±11 |
| hopper-me-v2 | 68±28 | 103 ± 17 | 74±22 | 94 ± 19 | 104±10 | 112 ± 1 |
| walker2d-r-v2 | 15±8 | 18 ± 16 | 18±10 | 65 ± 37 | 12±11 | 27 ± 42 |
| walker2d-mr-v2 | 81±17 | 108 ± 2 | 101±7 | 112 ± 12 | 70±35 | 109 ± 12 |
| walker2d-m-v2 | 88±7 | 96 ± 4 | 101±8 | 108 ± 5 | 92±26 | 102 ± 21 |
| walker2d-me-v2 | 113±0 | 118 ± 3 | 116 ± 1 | 111±4 | 111±1 | 120 ± 10 |
| **Locomotion total** | 810 | 933 | 890 | 1024 | 972 | 1081 |
| antmaze-mp-v2 | 82 ± 13 | 76±5 | 82±13 | 88 ± 11 | – | – |
| antmaze-md-v2 | 82±10 | 86 ± 5 | 90±14 | 98 ± 5 | – | – |
| antmaze-lp-v2 | 48±13 | 52 ± 18 | 54±19 | 56 ± 21 | – | – |
| antmaze-ld-v2 | 38±16 | 52 ± 22 | 38±16 | 42 ± 16 | – | – |
| **AntMaze total** | 250 | 266 | 264 | 284 | | |

[1] r: random, mr: medium-replay, m: medium, me: medium-expert, mp: medium-play, md: medium-diverse, lp: large-play, ld: large-diverse.

first paradigm, while APL follows the second, allowing us to test the generality of our algorithm.

We use the original papers' implementation for all three baselines. Every experiment starts with training a model only using the offline dataset, the same as in the original work. We note that, since APL did not conduct experiments on the AntMaze dataset, we also did not perform experiments on it.

**Settings** For IQL and PEX, we perform 1M update steps for offline pre-training and then 1M environment steps for online fine-tuning. For APL, we perform 1M pre-training steps and 0.1M fine-tuning steps to ensure consistency with the original paper. For our data augmentation method, the synthetic buffer size is set to 1M. The update frequency of $M$ is 10K in APL and 100K in IQL and PEX. The generated data ratio $r$ is set to $1/3$. Therefore, the percentage of online data, offline data and generated data is $1 : 1 : 1$. In the generated data, the ratio of generated online data to generated offline data is $8 : 2$. The above configurations keep the same across all tasks, datasets and methods. Some detailed CFDG parameters are included in Appendix A.2.

As shown in Table 1, the integration of CFDG outperforms all baselines. Using a diffusion model with classifier-free guidance to generate offline data and online data can surpass

the baseline algorithm on over 10 datasets in Locomotion tasks and 4 datasets in AntMaze tasks. After using CFDG for data augmentation, APL's performance improved by 11%, while both IQL and PEX achieved a 15% performance improvement.

### 4.2. Comparisons between CFDG and Model-based Methods

In addition to demonstrating the effectiveness of incorporating CFDG into standard O2O RL algorithms, we also aim to prove that CFDG outperforms current SOTA data augmentation methods. We compare CFDG with two model-based methods, SynthER (Lu et al., 2024) and EDIS (Liu et al., 2024). In O2O RL, SynthER directly uses a diffusion model to augment online data, while EDIS employs an energy-guided diffusion model to generate new data from offline data. The key difference between CFDG and these two methods is that CFDG separates offline data and online data into two distinct labels and uses a conditional diffusion model for generation. By accounting for the differences in the distributions of these two types of data and their distinct roles in O2O RL, performing data augmentation for each separately can significantly enhance the algorithm's performance. The base algorithm is IQL, and the results are shown in Figure 2 and Figure 3.

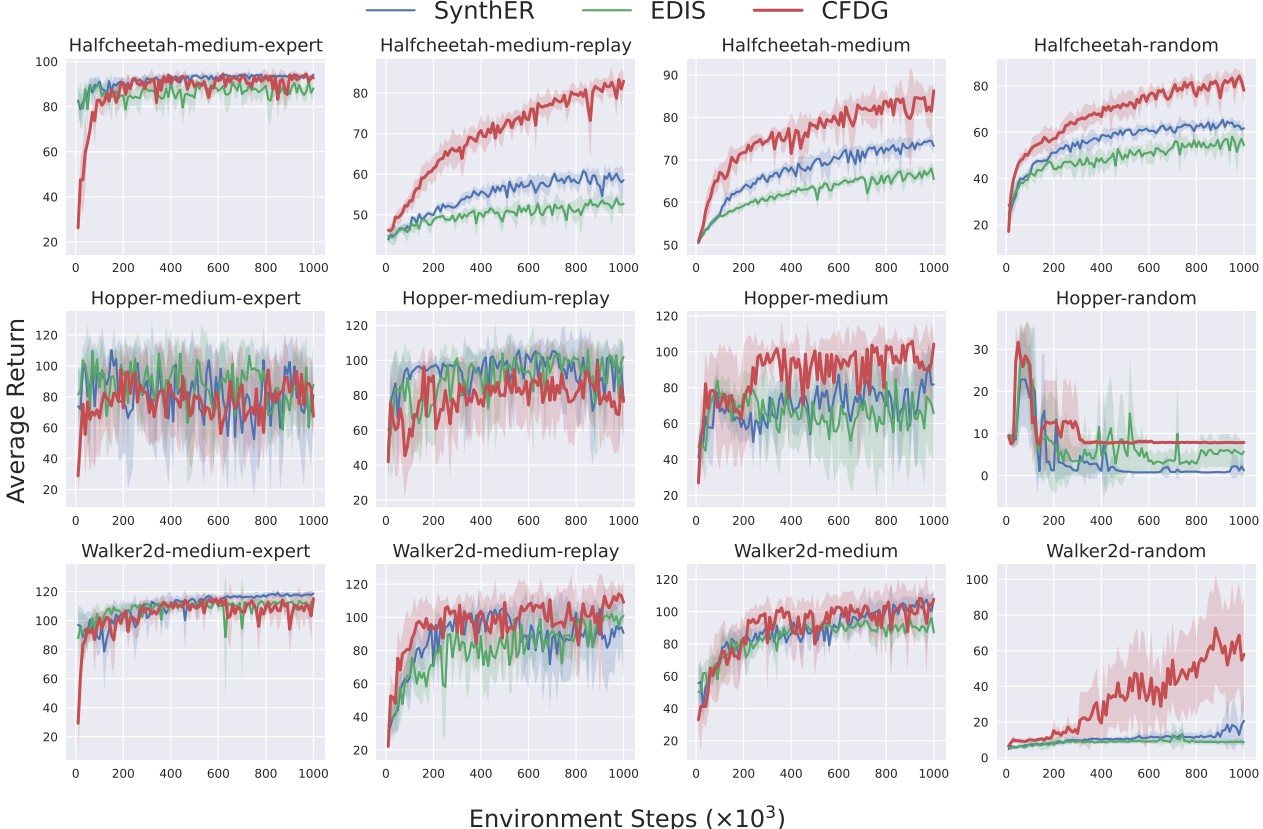

Figure 2: Learning curves of base algorithm augmented with model-based methods SynthER, EDIS and CFDG. Results are averaged over 5 random seeds.

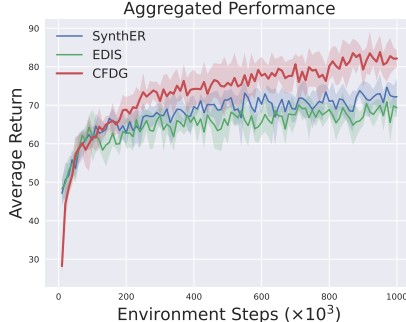

Figure 3: Aggregated learning curves of normalized return over 5 seeds on 12 Locomotion tasks.

Based on the experimental results, SynthER which using online data for data augmentation outperforms EDIS, which performs data augmentation based on offline data. It aligns with the intuition that online data are more aligned with the current policy. However, using CFDG to augment both online and offline data simultaneously further improves performance. This is particularly evident in the halfcheetah environment, where CFDG achieves a 15% performance improvement over previous model-based methods across 12

Locomotion tasks, shown in Figure 3.

To verify that the data generated by our method has higher quality and is better aligned with the current online policy, we analyzed the Jensen-Shannon (JS) divergence (Menéndez et al., 1997) between the data generated by EDIS and CFDG and the online data. We also compared the JS divergence between offline data and online data, as shown in Table 2. Our CFDG method generates data that is more closely aligned with the online data, which can be used for agent training, leading to improved performance.

Table 2: Divergence comparisons for generated data from (lower is better). Each result is the average score over 5 random seeds.

| Divergence | Offline | Generated EDIS | Generated CFDG |
|---|---|---|---|
| State | 0.69±0.01 | 0.56±0.02 | **0.50±0.01** |
| Action | 0.32±0.03 | 0.31±0.02 | **0.27±0.02** |
| Transition | 0.69±0.03 | 0.64±0.04 | **0.61±0.03** |

Compared to other model-based methods, CFDG can generate higher-quality data samples. In summary, performing data augmentation on both online and offline data allows

the agent to train on more comprehensive samples, leading to better performance.

### 4.3. Ablation Studies

The two main differences between the CFDG method and existing model-based approaches are: (i) It uses classifier-free guidance to steer the diffusion model in generating new data. (ii) It performs data augmentation on both offline data and online data. Therefore, we need to demonstrate that both components effectively enhance performance. We conduct an ablation study on the Locomotion tasks to compare the full CFDG method against its variants: one without classifier guidance and another with classifier guidance but without offline data augmentation. We evaluate performance across three environments: halfcheetah, hopper, and walker2d, averaging the performance across four tasks.

Table 3: **Ablation results of CFDG on Locomotion tasks.** All results are averaged over the four datasets and are assessed across 5 random seeds.

| Dataset | CFDG w/o guidance | CFDG w/o offline DA | CFDG |
|---|---|---|---|
| Halfcheetah | 80.65±2.43 | 81.22±1.78 | **84.44±2.15** |
| Hopper | 65.75±9.91 | 67.50±7.35 | **68.22±4.74** |
| Walker2d | 75.99±6.12 | 81.59±4.83 | **93.65±6.00** |
| **Average** | 74.13 | 76.77 | **82.10** |

As shown in Table 3, compared to standard diffusion, using classifier guidance improves the quality of data generation and enhances the performance of the agent after data augmentation. Building on classifier guidance, applying data augmentation to both offline and online data further improves the agent's performance. This highlights that both classifier guidance and data augmentation for both types of data are essential components of our CFDG method, contributing significantly to performance improvement.

### 5. Related Work

**Offline-to-online Reinforcement Learning**   Offline-to-online RL aims to improve suboptimal offline policies through online fine-tuning. Prior works usually improve the agent by adding a regularizer to mitigate the distribution shift problem or leverage both offline data and online data in online fine-tuning. IQL (Kostrikov et al., 2021) incorporates a weighted behavioral cloning step to enhance online policy improvement and is applicable in both online and offline-to-online scenarios. OFF2ON (Lee et al., 2022) employs a balanced replay scheme to address the distribution shift issue. It uses offline data by only selecting near-on-policy samples. However, practical scenarios may involve agents pretrained by various offline RL algorithms, highlighting the necessity for developing a generic offline-to-online RL framework.

Recent studies place a growing emphasis on adaptability. PEX (Zhang et al., 2023) freezes the pre-trained policy and initializes a random policy to enhance exploration. From a data-centric perspective, APL (Zheng et al., 2023) and SUNG (Guo et al., 2023) impose constraints exclusively on data from offline datasets and data with high uncertainty, respectively. Our method also focuses on data utilization, using data augmentation to expand the available data, thereby enabling more comprehensive learning and improving the agent's performance.

**Diffusion Models in RL**   Pearce et al. (Pearce et al., 2022) propose using a diffusion model to better imitate human behaviors due to their expressiveness and stability. Diffuser (Janner et al., 2022) applies a diffusion model as a trajectory generator, where the full trajectory of state-action pairs forms a single sample for the diffusion model. Additionally, a separate return model is trained to predict the cumulative rewards of each trajectory sample, and its guidance is incorporated into the reverse sampling stage. This approach is similar to Decision Transformer (Chen et al., 2021), which also learns a trajectory generator through GPT-2 with the help of the true trajectory returns. However, when used in online settings, sequence models can no longer predict actions from states autoregressively since the states are an outcome of the environment. Additionally, diffusion models are also used for data augmentation in RL. SynthER (Lu et al., 2024) directly uses a diffusion model to augment online data. EDIS (Liu et al., 2024) employs an energy-guided diffusion model to generate new data from offline data. Our method also uses diffusion models for data augmentation. However, considering the unique characteristics of the O2O RL settings and the differences between offline and online data, we utilize classifier-free guidance to generate new data.

### 6. Conclusion

In this paper, we analyze the distributions of offline and online data in the O2O RL setting and improve upon existing data augmentation methods by addressing their limitations. We use a diffusion model with classifier-free guidance to simultaneously augment both types of data. In the online phase, we input offline data and online data as two distinct labels into the diffusion model. With a single round of training, we can sample both types of data. Our method, CFDG, is simple and can be easily integrated with existing O2O RL algorithms, significantly boosting their performance while surpassing other data augmentation methods. Although our method has achieved superior performance, there is still room for future work. We can explore fine-tuning large pre-trained models and leveraging their generalization capabilities to synthesize new pixel-based environment data in RL.

## Acknowledgement

The corresponding author Shuai Li is sponsored by CCF-Tencent Open Research Fund.

## Impact Statement

This work presents a novel approach to data augmentation in Offline-to-Online Reinforcement Learning (O2O RL). By improving the alignment between generated data and online policy distributions, our method enhances training stability and overall agent performance while reducing computational overhead. The broader impact of this research includes advancing the efficiency of RL algorithms, potentially accelerating adoption in real-world applications such as robotics, autonomous systems, and decision-making frameworks that require fine-tuning with limited online interactions.

From an ethical perspective, our method does not inherently introduce bias but relies on the quality and representativeness of offline datasets. Future work should ensure diverse and unbiased data selection to mitigate unintended consequences. Additionally, while our approach optimizes RL performance, care must be taken in high-stakes applications to prevent over-reliance on synthetic data without proper validation.

Overall, this research contributes to the broader field of machine learning by improving sample efficiency in reinforcement learning while maintaining ethical considerations in data-driven decision-making.

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

# A. Appendix

## A.1. Adroit Results

To further validate the effectiveness and generalization of our method, we conduct additional experiments on the Adroit benchmark (relocate-human-v1, pen-human-v1, and door-human-v1), a suite of challenging and realistic continuous control tasks designed to test fine motor skills in a human-like hand. Our base algorithm in the Adroit setting is IQL, as EDIS is also evaluated on top of IQL. As shown in Table 4, our proposed method, CFDG, significantly outperforms both the base model and the prior state-of-the-art approach EDIS across all three tasks.

Table 4: Comparisons between CFDG and EDIS on Adroit tasks. Each result is the average score over 5 random seeds.

| Dataset | Base | EDIS | CFDG |
|---|---|---|---|
| relocate-human-v1 | 0.4±0.4 | 0.2±0.6 | **1.5±1.8** |
| pen-human-v1 | 72.2±64.0 | 73.4±10.3 | **96.8±59.4** |
| door-human-v1 | 4.5±7.9 | 6.2±3.2 | **31.7±6.5** |
| **Average** | 25.7 | 26.6 | **43.3** |

Our results show that CFDG achieves over 50% improvement over baselines and the model-based EDIS approach on Adroit tasks. These results demonstrate that CFDG is not only effective in standard benchmark settings but also robust in more realistic and difficult environments, further supporting its applicability to real-world scenarios.

## A.2. Details and Hyperparameters of CFDG

We use the PyTorch implementation of IQL and PEX from https://github.com/Haichao-Zhang/PEX, and the implementation of APL from https://github.com/zhan0903/APL0 and primarily followed the authors' recommended parameters. The hyperparameters used in our CFDG module are detailed in Table 5:

Table 5: Hyperparameters and their values in CFDG.

| Hyperparameter | Value |
|---|---|
| Denoising Network | Residual MLP |
| Denoising Network Depth | 6 layers |
| Denoising Steps | 128 steps |
| Denoising Network Learning Rate | $3 \times 10^{-4}$ |
| Denoising Network Hidden Dimension | 1024 units |
| Denoising Network Batch Size | 256 samples |
| Denoising Network Activation Function | ReLU |
| Denoising Network Optimizer | Adam |
| Learning Rate Schedule | Cosine Annealing |
| Training Epochs | 100K epochs |
| Training Interval Environment Step | 10K steps (APL), 100K steps (IQL & PEX) |

The formulation of diffusion we use in our paper is the Elucidated Diffusion Model (Karras et al., 2022). We parametrize the denoising network $D_\theta$ as an MLP with skip connections from the previous layer. The base size of the network uses a width of 1024 and depth of 6. We use a batch size of 256 for all tasks. For the diffusion sampling process, we use the stochastic SDE sampler (Karras et al., 2022) with the default hyperparameters used for the ImageNet. We use a higher number of diffusion timesteps at 128 for improved sample fidelity. We use the implementation at https://github.com/lucidrains/denoising-diffusion-pytorch.

## A.3. Computational Resources

We train CFDG integrated with base algorithms on an NVIDIA RTX 2080Ti, with approximately 23 hours required for 10K fine-tuning on MuJoCo Locomotion tasks in APL, while 16 hours for 100K fine-tuning in IQL & PEX. The detailed computational consumption is shown in Table 6. As pointed out in (Karras et al., 2022), the sampling time is faster than prior diffusion designs, which is much shorter compared with training. The introduction of the diffusion model does indeed entail an inevitable increase in computational and time costs. However, this tradeoff between improved performance and

higher computational cost is a common consideration in diffusion model research. In our future work, we aim to further refine and optimize the extra costs.

Table 6: Computational consumption of different algorithms.

| Algorithm | Online phase training time | Maximal GPU memory |
|---|---|---|
| APL | 20h | 2G |
| APL-CFDG | 23h | 3G |
| PEX | 14h | 2G |
| PEX-CFDG | 17h | 3G |
| IQL | 12h | 2G |
| IQL-CFDG | 15h | 3G |

