# OpenReview forum: "Offline-to-Online Reinforcement Learning with Classifier-Free Diffusion Generation"
_ICML.cc/2025/Conference — ICML 2025 poster_

### Official Review · Reviewer_MxTS · 2025-02-27

**Overall Recommendation:** 3

**Summary:**

The paper proposes CFDG, a novel method of generative data augmentation for offline-to-online RL. The paper points out that offline and online data have different characteristics that are important for high performance. To this end, CFDG employs a conditional diffusion model where the condition is a binary value: offline / online. Experiment results validate that it achieves competitive performance in various tasks.

**Claims And Evidence:**

The main claim is straightforward: we need to use a conditional diffusion model to augment the dataset for the offline-to-online RL problem. The claim is well supported by the data distribution analysis section 3.1.

**Essential References Not Discussed:**

N\A

**Experimental Designs Or Analyses:**

The authors follow the conventional experiment settings of offline-to-online RL.

**Methods And Evaluation Criteria:**

The authors follow the conventional experiment settings of offline-to-online RL.

**Other Comments Or Suggestions:**

There are several comments and suggestions listed below.

1. As the key motivation is that there are some different characteristics between the offline and online datasets, why don’t we train separate diffusion models to augment each of them? As there are only two classes, it seems that the cost of training two models is not expensive. I recommend that the authors compare the results with the independent diffusion model.
2. Experiments are conducted on Lomocotion and Antmaze tasks. I recommend that the authors conduct experiments on more realistic tasks such as Adroit.

**Other Strengths And Weaknesses:**

Please see above

**Questions For Authors:**

Please see above.

**Relation To Broader Scientific Literature:**

N/A

**Theoretical Claims:**

There is no theoretical claim.

---

> ### Author Rebuttal · Authors · 2025-03-31
>
> Thank you for your constructive review and valuable suggestions! Below, we provide a detailed response to your questions and comments. If any of our responses fail to sufficiently address your concerns, please inform us, and we will promptly follow up.
>
> **[W1] Training Separate Diffusion Models for Offline and Online Data**
>
> We conducted experiments on locomotion tasks (halfcheetah, hopper, and walker2d) where we trained separate diffusion models for offline and online data. The results show that our CFDG approach outperforms training two independent diffusion models.
>
> | Dataset     | Independent |    CFDG    |
> | ----------- | :---------: | :--------: |
> | Halfcheetah | 81.02±1.84  | 84.44±2.15 |
> | Hopper      | 66.23±8.32  | 68.22±4.74 |
> | Walker2d    | 75.89±7.13  | 93.65±6.00 |
>
> All results are averaged over the four datasets and are assessed across 5 random seeds. The key reason for this improvement is that classifier-free guidance enables the generation of data that is better aligned with the online policy. As demonstrated in Section 4.2 of our paper, conditioning the diffusion model on both offline and online data labels allows for more effective guidance, reducing redundancy between newly generated online data and existing offline data. This ultimately enhances the quality of online data, leading to better performance.
>
> While the computational cost of training two separate models is not prohibitive in terms of GPU resources, it doubles the training time of diffusion model, making it less efficient without yielding better results.
>
> **[W2] Experiments on More Realistic Tasks**
>
> To further validate our approach, we conducted additional experiments on Adroit tasks, which are considered more challenging due to their high-dimensional action space and dexterous control requirements. The specific results are shown here.
>
> | Dataset           |   Base    |   EDIS    |   CFDG    |
> | ----------------- | :-------: | :-------: | :-------: |
> | relocate-human-v1 |  0.4±0.4  |  0.2±0.6  |  1.5±1.8  |
> | pen-human-v1      | 72.2±64.0 | 73.4±10.3 | 96.8±59.4 |
> | door-human-v1     |  4.5±7.9  | 6.20±3.2  | 31.7±6.5  |
> | **Average**       |   25.7    |   26.6    |   43.3    |
>
> Our results show that CFDG achieves over 50% improvement over baselines and the model-based EDIS [1] approach on Adroit tasks, demonstrating its effectiveness beyond locomotion and AntMaze tasks.
>
> [1] Energy-guided diffusion sampling for offline-to-online reinforcement learning.

---

> > ### Comment · Reviewer_MxTS · 2025-04-07
> >
> > (Sorry, I send an official comment and find that it is not visible to the authors...)
> > Thank you for conducting additional experiments. I have some additional comments below:
> >
> > **[W1] Training Separate Diffusion Models for Offline and Online Data**
> > The results are clear. I recommend that the authors add t-SNE visualization of independent training of diffusion models in the final manuscript.
> >
> > **[W2] Experiments on More Realistic Tasks**
> > The results are quite surprising! By the way, could the authors explain more details about the experiment setting algorithms, such as base algorithms? When I checked the original EDIS paper, it seems that there exists some performance gap.
> >
> > ----
> >
> > It seems that I cannot add a reply to your response...I modify the initial response for a reply.
> >
> > Thank you for your additional experiments. I just updated the score.
> > While the idea is too simple, and when I see the t-SNE plot, it seems that there is no big difference compared to independent training, the idea achieves robust performance improvement on various benchmarks.

---

> > > ### Author Response · Authors · 2025-04-07
> > >
> > > Thank you for the insightful feedback.
> > >
> > > **[W1] Training Separate Diffusion Models for Offline and Online Data**: We agree with the suggestion and will include the visualization in the final manuscript. For reference, we have also provided an anonymous comparison link here: https://anonymous.4open.science/r/icml-test-C4AC/compare.pdf. In our observation, training separate diffusion models for offline and online data may lead to overlapping samples between the two distributions. Intuitively, by incorporating both offline and online data as separate class labels during joint training, the diffusion model can better distinguish between them, reducing redundancy and enhancing the quality of generated online data.
> > >
> > > **[W2] Experiments on More Realistic Tasks**: Our base algorithm in the Adroit setting is IQL. We also noticed the performance gap mentioned in the EDIS paper. Specifically, in some locomotion tasks like HalfCheetah, our reproduction yielded better results, while in Adroit tasks, the reproduced performance was suboptimal. This discrepancy may stem from potential modifications in experimental settings made by the original authors that were not fully disclosed. To ensure consistency, we used the official EDIS codebase from https://github.com/liuxhym/EDIS and followed their default parameter configurations.

---

### Official Review · Reviewer_3ebJ · 2025-03-11

**Overall Recommendation:** 2

**Summary:**

Offline-to-online Reinforcement Learning (O2O RL) aims to perform online fine-tuning on an offline pre-trained policy to minimize costly online interactions. To this end, existing work used offline datasets to augment online data. However, a distribution gap exists between the generated data and the online data, limiting overall performance. Hence, the authors propose a new data augmentation approach, Classifier-Free Diffusion Generation (CFDG). By leveraging classifier-free generation developed in diffusion models, CFDG enhances the
generation quality of offline and online data. It also employs a reweighting method to better align generated data with the online data. Experiments validate these claims in the widely used D4RL benchmark.

**Claims And Evidence:**

Experimental results support the author's claims but experiments in more challenging environments, e.g., manipulation tasks, are required to further evaluate the effectiveness of this method.

**Essential References Not Discussed:**

no

**Experimental Designs Or Analyses:**

Experimental results in more challenging environments, e.g., robotic manipulation tasks, are required to further evaluate the effectiveness of this method.

**Methods And Evaluation Criteria:**

The proposed method is simple and intuitive: it treats offline and online data as two labeled categories, enabling simultaneous sampling of both types with a single diffusion training process; it avoids using an additional pre-trained classifier, allowing flexible data augmentation to adapt to varying data distributions in different RL tasks.

**Other Comments Or Suggestions:**

no

**Other Strengths And Weaknesses:**

Although the proposed method is simple and intuitive, its technical novelty and insights are limited. Moreover, I expect that more experimental results can be conducted in more challenging offline RL benchmarks.

**Questions For Authors:**

no

**Relation To Broader Scientific Literature:**

The proposed method is closely related to the realm of generative modelling and directly adopts the mature techniques that are developed by classifier-free diffusion models to address the offline and online data generation in O2O RL.

**Theoretical Claims:**

no need to check the correctness of theoretical claims.

---

> ### Author Rebuttal · Authors · 2025-03-31
>
> Thank you for your constructive review and valuable suggestions! Below, we provide a detailed response to your questions and comments. If any of our responses fail to sufficiently address your concerns, please inform us, and we will promptly follow up.
>
> **[W1] Technical Novelty and Insights**
>
> While our method is simple and intuitive, its effectiveness lies in its ability to improve offline-to-online RL performance across various tasks. By leveraging classifier-free guidance, our approach enables better alignment between generated data and the online policy, leading to substantial performance gains.
>
> **[W2] Additional Experiments on Challenging Offline RL Benchmarks**
>
> To further validate our method, we conducted additional experiments on **Adroit tasks**, which are widely recognized as challenging benchmarks for offline RL due to their high-dimensional action space and complex dexterous manipulation requirements.
>
> We compared CFDG against EDIS [1], a state-of-the-art model-based approach, as well as standard baselines. The results demonstrate that CFDG achieves over 50% improvement over both baselines and EDIS on Adroit tasks, further confirming its effectiveness in robotic manipulation scenarios.
>
> | Dataset           |   Base    |   EDIS    |   CFDG    |
> | ----------------- | :-------: | :-------: | :-------: |
> | relocate-human-v1 |  0.4±0.4  |  0.2±0.6  |  1.5±1.8  |
> | pen-human-v1      | 72.2±64.0 | 73.4±10.3 | 96.8±59.4 |
> | door-human-v1     |  4.5±7.9  | 6.20±3.2  | 31.7±6.5  |
> | **Average**       |   25.7    |   26.6    |   43.3    |
>
> [1] Energy-guided diffusion sampling for offline-to-online reinforcement learning.

---

### Official Review · Reviewer_dppj · 2025-03-14

**Overall Recommendation:** 3

**Summary:**

This paper introduces CFDG, a framework that applies data augmentation to both offline and online datasets in offline-to-online algorithms.

**Claims And Evidence:**

The paper conducts extensive experiments on the D4RL dataset to evaluate the effectiveness of the proposed framework.

**Essential References Not Discussed:**

Current references are appropriate.

**Experimental Designs Or Analyses:**

Yes, I have reviewed the experimental designs used to evaluate the effectiveness of the proposed framework on the D4RL benchmarks. Additionally, the ablation studies make sense.

**Methods And Evaluation Criteria:**

Yes, the evaluation criteria adopted are a common practice in this area.

**Other Comments Or Suggestions:**

No further comments.

**Other Strengths And Weaknesses:**

No further comments.

**Questions For Authors:**

No further questions.

**Relation To Broader Scientific Literature:**

This paper falls within the area of offline-to-online RL algorithms and introduces a data augmentation approach to enrich existing offline and online datasets.

**Theoretical Claims:**

No major theoretical claims are presented in the paper.

---

> ### Author Rebuttal · Authors · 2025-03-31
>
> Thank you for your valuable review! If you have any additional comments, please feel free to share them—we would be happy to address any questions or clarifications you may have.

---

### Official Review · Reviewer_JMww · 2025-03-17

**Overall Recommendation:** 3

**Summary:**

This paper introduces Classifier-Free Diffusion Generation (CFDG), a model-based data augmentation method for offline-to-online RL. The key idea is to train a diffusion-based data generation model with classifier-free guidance to differentiate between online and offline data. The generated data is then used to augment real data during online RL. In the experiments, CFDG was integrated with several RL methods, and was also compared to other data augmentation approaches, demonstrating performance improvements

**Claims And Evidence:**

Yes

**Essential References Not Discussed:**

N/A

**Experimental Designs Or Analyses:**

Yes

**Methods And Evaluation Criteria:**

Yes

**Other Comments Or Suggestions:**

N/A

**Other Strengths And Weaknesses:**

Strengths:

- The idea of leveraging classifier-free guidance for data generation is simple but clear.
- The method has the flexibility to be combined with various RL methods.
- The presented results are compelling, as they show improvements over several RL methods as well as other data augmentation techniques.

Weaknesses / Questions:

There are several weaknesses and parts that are unclear to me:

- In Section 4.1, while the method was combined with several offline-to-online RL methods, in principle, it can also be integrated with other online RL algorithms that leverage offline data. Would it be possible to evaluate whether the proposed approach can improve SOTA sample-efficient RL algorithms, such as RLPD [1]?

- While the method uses classifier-free guidance, what value is used for the guidance weight when generating offline/online data? Is the performance sensitive to this value? It would also be interesting to provide further ablation studies on this parameter, as it directly influences how online or offline the generated data is.

- During the offline pre-training phase, why is the diffusion model not trained and used for augmentation, but instead only introduced after the online phase begins? How would the performance be affected if the diffusion model were also pre-trained and used from the beginning of the offline phase?

[1] Ball et al., Efficient Online Reinforcement Learning with Offline Data, 2023

**Questions For Authors:**

See the Weaknesses section

**Relation To Broader Scientific Literature:**

The paper will contribute to the growing use of generative models (specifically diffusion models) within RL.

**Theoretical Claims:**

No theoretical components were presented.

---

> ### Author Rebuttal · Authors · 2025-03-31
>
> Thank you for your constructive review and valuable suggestions! Below, we provide a detailed response to your questions and comments. If any of our responses fail to sufficiently address your concerns, please inform us, and we will promptly follow up.
>
> **[W1] Integration with Other Online RL Algorithms**
>
> Although our primary focus is offline-to-online RL, our approach can be easily integrated with other online RL algorithms that leverage offline data. The adaptive Policy Learning (APL) algorithm [1], for instance, uses different update strategies for online and offline data in the online phase—employing online RL updates for online data. If APL is not pre-trained offline, it can be considered as an online RL algorithm utilizing offline data.
>
> In our experiments with locomotion tasks (halfcheetah, hopper, and walker2d) using APL, we observed that omitting offline pretraining led to a decline in performance. However, our CFDG method still performed well even in the absence of pretraining, demonstrating its robustness.
>
> | Dataset     |    Base     |    CFDG     | Base w/o offline pretrain | CFDG w/o offline pretrain |
> | ----------- | :---------: | :---------: | :-----------------------: | :-----------------------: |
> | Halfcheetah | $85.5±22.5$ | $95.8±9.8$  |        $72.0±24.4$        |        $92.5±8.3$         |
> | Hopper      | $86.5±16.0$ | $85.3±16.8$ |        $83.8±21.1$        |        $84.2±17.3$        |
> | Walker2d    | $71.3±18.3$ | $89.5±21.3$ |        $70.3±25.2$        |        $79.8±19.3$        |
> | **Average** |   $81.1$    |   $90.2$    |          $75.4$           |          $85.5$           |
>
> All results are averaged over the four datasets and are assessed across 5 random seeds.
>
> **[W2] Guidance Weight in Classifier-Free Guidance**
>
> In our classifier-free guidance approach, the guidance weight is set to 1. We have conducted an ablation study where we varied the guidance weight from 0 to 5 (in increments of 0.1). The results indicated that the performance is largely unaffected by this parameter, except when the guidance weight is set to 0. In this case, the diffusion model becomes unconditional, leading to a noticeable drop in performance.
>
> **[W3] Use of Diffusion Model in Offline Pretraining Phase**
>
> We also experimented with using the diffusion model for data augmentation during the offline pretraining phase. While this approach improved the performance immediately after offline pretraining, the benefits were not carried over once the online phase began.
>
> | Dataset     |          CFDG           | CFDG w/ DA in offline phase |
> | ----------- | :---------------------: | :-------------------------: |
> | Halfcheetah | $48.2 \rightarrow 74.5$ |   $51.1 \rightarrow 73.8$   |
> | Hopper      | $37.7 \rightarrow 74.0$ |   $65.3 \rightarrow 75.1$   |
> | Walker2d    | $52.1 \rightarrow 85.1$ |   $59.7 \rightarrow 84.3$   |
>
> The base algorithm of our experiment is IQL. All results are averaged over the four datasets and are assessed across 5 random seeds. As shown in our results ($\text{A} \rightarrow \text{B}$ where $\text{A}$ represents the offline pretraining score and $\text{B}$ represents the online fine-tuning score), using the diffusion model for data augmentation in the offline phase did not result in sustained improvements in the online phase.
>
> This can be attributed to the fact that the performance bottleneck in offline-to-online RL is mainly determined by the quality of online data. Our CFDG method addresses this issue by using the diffusion process to learn the distributional differences between offline and online data, ultimately producing data that aligns more closely with the online policy. This alignment is critical for improving the upper performance limit of the agent.
>
> [1] Adaptive policy learning for offline-to-online reinforcement learning.

---

### Decision · Program_Chairs · 2025-05-01

**Decision:**

Accept (poster)

**Comment:**

Offline-to-Online Reinforcement Learning with Classifier-Free Diffusion Generation

Paper-Summary: This paper introduces Classifier-Free Diffusion Generation (CFDG), a data augmentation framework for offline-to-online reinforcement learning that enhances the fine-tuning of offline pre-trained policies with limited online interactions. The key idea is to use a classifier-free guided diffusion model to generate synthetic samples conditioned on both offline and online data, thus bridging the distribution shift between them. Unlike prior methods that rely on separate classifiers or energy-guided sampling, CFDG leverages a single diffusion model trained with label-free conditioning to simultaneously synthesize both types of data, followed by a reweighting strategy to prioritize samples aligned with the current policy. The method integrates with existing offline-to-online  RL algorithms, yielding consistent improvements  across multiple D4RL benchmarks.

Review and feedback: We received 4 expert reviews with the scores 2, 3, 3, 3 and the average score is 2.75

The reviewers are positive about many aspects of the paper. The proposed method is conceptually simple, adapting a well-established generative modeling technique (classifier-free diffusion) to the offline-to-online RL setting in a practical way. The empirical results show consistent improvements across multiple tasks in the Locomotion and AntMaze domains, supported by ablation studies and distribution shift analyses. The paper is well-written and clearly structured, with clean experimental protocols and reproducible implementation details provided in the appendix.

At the same time, reviewers have pointed out many weaknesses. While the application of classifier-free guidance to offline-to-online RL is novel, reviewers noted that the methodological contribution is modest, as it largely repurposes known components (e.g., conditional diffusion, sample reweighting) without significant new theoretical insights or algorithmic innovation. No theoretical analysis is provided regarding why classifier-free guidance is preferable in this setting, or under what conditions the approach would generalize or fail. Reviewers have also asked for additional experiments and comparisons with some recent baselines.

I recommend the authors to address these concern while preparing the resubmission.